# Overexpression of *β-Ketoacyl CoA Synthase 2B.1* from *Chenopodium quinoa* Promotes Suberin Monomers’ Production and Salt Tolerance in *Arabidopsis thaliana*

**DOI:** 10.3390/ijms232113204

**Published:** 2022-10-30

**Authors:** Faheem Tariq, Shuangshuang Zhao, Naveed Ahmad, Pingping Wang, Qun Shao, Changle Ma, Xianpeng Yang

**Affiliations:** 1Shandong Provincial Key Laboratory of Plant Stress, College of Life Sciences, Shandong Normal University, Jinan 250014, China; 2Institute of Crop and Germplasm Resource (Institute of Biotechnology), Shandong Academy of Agricultural Sciences, Jinan 250100, China

**Keywords:** *Chenopodium quinoa*, *β-Ketoacyl CoA Synthase2B.1*, very-long-chain fatty acids, suberin monomers, lateral root, salt tolerance

## Abstract

Very-long-chain fatty acids (VLCFAs) are precursors for the synthesis of various lipids, such as triacylglycerols, sphingolipids, cuticular waxes, and suberin monomers, which play important roles in plant growth and stress responses. However, the underlying molecular mechanism regulating VLCFAs’ biosynthesis in quinoa (*Chenopodium quinoa* Willd.) remains unclear. In this study, we identified and functionally characterized putative 3-ketoacyl-CoA synthases (KCSs) from quinoa. Among these *KCS* genes, *CqKCS2B.1* showed high transcript levels in the root tissues and these were rapidly induced by salt stress. *CqKCS2B.1* was localized to the endoplasmic reticulum. Overexpression of *CqKCS2B.1* in Arabidopsis resulted in significantly longer primary roots and more lateral roots. Ectopic expression of *CqKCS2B.1* in *Arabidopsis* promoted the accumulation of suberin monomers. The occurrence of VLCFAs with C22–C24 chain lengths in the overexpression lines suggested that *CqKCS2B.1* plays an important role in the elongation of VLCFAs from C20 to C24. The transgenic lines of overexpressed *CqKCS2B.1* showed increased salt tolerance, as indicated by an increased germination rate and improved plant growth and survival under salt stress. These findings highlight the significant role of *CqKCS2B.1* in VLCFAs’ production, thereby regulating suberin biosynthesis and responses to salt stress. *CqKCS2B.1* could be utilized as a candidate gene locus to breed superior, stress-tolerant quinoa cultivars.

## 1. Introduction

Very-long-chain fatty acids (VLCFAs) are fatty acids with more than 18 acyl carbons in their backbone chain. These fatty acids are structurally and functionally diverse, and their activity depends on their chain length, degree of unsaturation, associated lipids and polar head type. The VLCFAs are present in the form of triacylglycerol in seeds and cuticle waxes that are deposited on the primary surfaces of plants [1]. A modified form of VLCFAs is also present in suberin [2]. The VLCFAs have a variety of functions in plant growth and development, cell-to-cell transport, and hormone signal regulation. For example, in cotton, a VLCFA–ethylene pathway controls cotton fiber length [3]. In *Arabidopsis thaliana,* VLCFA synthesis in the epidermis inhibits excessive cytokinin production and cell proliferation, and controls shoot growth at the apices by inhibiting cytokinin biosynthesis at the shoot tips [4]. During cytokinesis and cell differentiation, VLCFA-containing phospholipids play a key role in endomembrane dynamics [5]. As well as their roles in growth and development, VLCFAs are actively involved in conferring tolerance to a wide range of abiotic and biotic stresses [6,7]. For example, under salt stress, the contents of saturated and unsaturated VLCFAs with chain lengths of C20–C26 increase in *Chenopodium album* L., with a greater increase in the root than in the leaf cuticle [8]. In *Arabidopsis*, the ectopic expression of *CsKCS6* was found to increase VLCFA accumulation in the cuticle wax on the leaves, thereby enhancing tolerance to drought and saline stress [9]. The adaptive response to salt stress in halophytic plants includes increased saturation and length of fatty acids (FAs), resulting in a rigid membrane and decreased salt permeability [10]. 

In plants, VLCFAs are synthesized in the endoplasmic reticulum. The synthesized C16–C18 acyl CoAs in the plastid are catalyzed by the fatty acid elongase (FAE) complex through a cyclic reaction. In every cycle, the carbon chain length of acyl CoA is extended by two carbons. The FAE complex is generally composed of β-ketoacyl CoA synthase (KCSs), β-ketoacyl CoA reductase (KCR) [11], 3-hydroxyacyl CoA dehydratase (HCD) [12], and enoyl CoA reductase (ECR) [13]. Due of its substrate specificities, KCS is the rate-limiting enzyme in the FAE [14]. Arabidopsis has 21 *KCS* genes in its genome. *AtKCS2/DAISY* and *AtKCS20* elongate fatty acids from C20 to C22 and are associated with the regulation of root cork and the waxy cuticle of the plant [15]. Similarly, *AtKCS6/AtCUT1/AtCER6* is involved in the biosynthesis of VLCFAs with a chain length longer than C24 that localizes in the pollen epidermis and waxy stems [16]. Another member of the KCS family, *AtKCS10/AtFDH*, is a key regulator of epidermal wax synthesis and is expressed in the flower and young leaf [17]. Several studies have focused on *KCS* genes and their wide range of functions in diverse plant species. In the liverwort *Marchantia polymorpha, MpFAE2* encodes a KCS that catalyzes the VLCFA elongation of C18 to C22 [18]. Transgenic *Arabidopsis* overexpressing *AhKCS1* and *AhKCS28* from peanut showed increased VLCFA contents, especially saturated VLCFAs, in the seeds [19]. In citrus, *CsKCS2* and *CsKCS11* are actively involved in fruit cuticle wax accumulation during ripening [20]. In rice, *WSL4* is involved in the elongation of C22 to C24 and beyond, which requires the participation of *OsCER2* and regulates the synthesis of cuticle wax on the leaves [21]. Although *KCS* genes from many species have been cloned and characterized, very little is known about the function of the *KCS* gene family in halophytes. 

*Chenopodium quinoa* (2n = 4x = 36) is one of the oldest cultivated crops, being domesticated in South America as a staple food more than 7000 years ago. Quinoa is a pseudo-cereal facultative halophytic crop belonging to the Amaranthaceae family that is tolerant to a wide range of environmental stresses [22,23,24]. Because of quinoa’s nutritional importance, the demand for processed products has substantially increased [25]. Quinoa crops have an exceptional nutritional balance of protein, carbohydrates, and starch, are a good source of high-linoleate oil and vitamins, and lack gluten [22]. The importance of quinoa as an emerging crop is highlighted by the fact that the United Nations (UN) declared 2013 the “International Year of Quinoa” [26,27]. The results of molecular, cytogenic, and genetic analyses indicate an allotetraploid mode of origin for quinoa [28,29]. The current tetraploid quinoa is thought to be a result of the hybridization of ancestral A and B genomes of diploid species. Salt stress greatly affects plant growth and development. It has been estimated that 50% of arable land will be lost as a result of salinization by 2050 [30]. Quinoa is strongly tolerant to salt and drought stress, it is considered to be a model halophytic crop [31]. That can tolerate a high level of salt stress ranging from 150 mM to 750 mM NaCl [32]. 

In this study, we comprehensively identified the gene encoding KCSs in the *Chenopodium quinoa* genome. Then, we analyzed the expression of two homologs of *Arabidopsis AtKCS2* (*CqKCS2A.1* and *CqKCS2B.1*) in quinoa seedlings under salt stress. To reveal the biological function of CqKCS2B.1, we analyzed its subcellular localization and generated *Arabidopsis* plants by overexpressing its encoding gene. The results of this study provide valuable insight into how halophyte quinoa tolerates saline conditions.

## 2. Results

### 2.1. Phylogenetic Analysis of Quinoa CqKCS 

We searched for members of the *CqKCS* gene family in the quinoa genome using sequences of 21 KCSs from *A. thaliana* as the search queries (Appendix A). We identified 33 members of the KCS protein family in quinoa. All of them had two conserved domains of FAE-type KCSs; FAE1_CUT1_RppA and ACP_syn_III_C (Appendix A). Analyses of the CqKCS gene family revealed a duplication event, followed by sub-functionalization and neo-functionalization to generate more complex functions. We identified 16 paralogous genes in the *CqKCS* gene family (Appendix A). To further elucidate the evolutionary relationships of *CqKCS* genes, a phylogenetic tree was constructed using the 33 KCSs (FAE-type) from quinoa and 21 Arabidopsis protein sequences with MEGA11. The KCS proteins formed seven groups, I–VII. All the groups contained members of the CqKCS family, but Group V lacked AtKCS.

In the phylogenetic tree, the CqKCS family was separated into seven subfamilies (Figure 1). In Group I, CqKCSs were grouped with *AtKCS3*, *AtKCS12*, *AtKCS19*, *AtKCS7*, and *AtKCS21*, whose functions are unclear. Group II contained CqKCSs and *AtKCS5* and AtKCS6/CER6, which are involved in the synthesis of cuticular wax and pollen lipids [33,34]. In Group III, CqKCSs were grouped with *AtKCS1*, *AtKCS13*, and *AtKCS14*, which are involved in wax biosynthesis and signaling to limit regeneration by restricting pericycle callus formation [35,36]. In Group IV, CqKCSs were grouped with *AtKCS2/DAISY* and *AtKCS20*, which are associated with the biosynthesis of cuticular wax and suberin monomers [37]. Group V contained CqKCS genes but no AtKCSs, suggesting that these proteins are specific to quinoa or have substantially different functions from those in *Arabidopsis*. Group VI contained CqKCSs and *AtKCS4*, *AtKCS9*, *AtKCS8*, *AtKCS16*, *AtKCS17*, and *AtKCS18/FAE1*, which are primarily involved in the synthesis of tetracosanoic acid as a precursor for cuticle wax, suberin, and phospholipids, as well as seed fatty acid synthesis [38,39,40,41]. Group VII contained CqKCSs as well as *AtKCS10/AtFDH* and AtKCS15, which are involved in lipid accumulation during pollen stigma interaction and VLCFA production in epidermal cells [17,42,43]. *CqKCS2A.1* and *CqKCS2B.1* were grouped with *AtKCS2* and AtKCS20 in the phylogenetic analysis, suggesting that they may have similar functions. Therefore, we selected *CqKCS2A.1* and *CqKCS2B.1* for further analyses.

### 2.2. Expression Pattern of CqKCS2 in C. Quinoa 

We conducted RT-qPCR analyses to detect the transcript levels of *CqKCS2A.1* and *CqKCS2B.1* in different organs of quinoa. Due to the highly homologous sequences of *CqKCS2A.1* and *CqKCS2B.1* (i.e., 95.69%), it was difficult to distinguish between the two genes with specific RT-qPCR primers (Appendix A). Therefore, we detected the total *CqKCS2* transcript level (*CqKCS2A.1* and *CqKCS2B.1*) in different tissues of quinoa plants (Figure 2a). Transcripts of *CqKCS2* were detected in almost all organs, with the highest levels in roots. We then designed specific semi-RT-PCR primers for these two genes and confirmed their specificity using *35S::CqKCS2A.1-YFP* and *35S::CqKCS2B.1-YFP* vectors as templates (Appendix A). The transcript levels of both CqKCS2A.1 and CqKCS2B.1 were high in the roots and young leaves (Appendix A). Considering the high salt tolerance of quinoa, we determined the transcript levels of these genes in quinoa seedlings under salt stress. For these analyses, 3-day-old quinoa seedlings were transferred to a solid half-strength Murashige and Skoog medium (½ MS) containing 300 mM NaCl. The total RNA was extracted from seedlings at 0, 1, 3, 6, 12, and 24 h (Figure 2b) of salt stress treatment. Then, semi-RT-PCR was performed with *CqTUB6* as a reference gene and *CqCMO*, whose expression is induced by salt stress, as a positive control [44]. The transcript levels of *CqKCS2A.1* and especially *CqKCS2B.1* rapidly accumulated until 3 h of salt stress, and then decreased as the salt treatment extended. These findings showed that both *CqKCS2* genes (*CqKCS2A.1* and *CqKCS2B.1*) are involved in the response to salt stress. *CqKCS2B.1* was selected for further functional analyses because of its higher expression in response to salt stress. 

### 2.3. Localization of CqKCS2B.1 in the Endoplasmic Reticulum 

The VLCFAs are synthesized in the endoplasmic reticulum (ER), suggesting that *CqKCS2B.1* might be localized to this site. To confirm this, the *CqKCS2B.1* Coding sequence (CDs) sequence was C-terminally fused with YFP (*Pro35S::CqKCS2B.1-YFP*) and co-expressed with an ER-specific marker (CD3-959) in tobacco leaves. When the leaves were observed under a confocal microscope, the *CqKCS2B.1*-YFP was merged with the ER-mCherry signal, and both were localized to a network-like structure. This result confirmed that CqKCS2B.1 is an ER-localized protein (Figure 3). The ER-localization of CqKCS2A.1 was also investigated by transiently expressing *Pro35S::CqKCS2A.1-YFP* in tobacco leaves (Appendix A).

### 2.4. Ectopic Expression of CqKCS2B.1 Promotes Lateral Root Development in Arabidopsis

To further investigate the function of *CqKCS2B.1*, the *Pro35S::CqKCS2B.1-YFP* construct was stably transformed into Arabidopsis (Col-0). A total of 13 transgenic lines were obtained. Two *CqKCS2B.1* overexpressing lines, OE#5 and OE#6, were selected for further analyses because of their high expression levels of *CqKCS2B.1* as detected by Western blotting with anti-GFP antibodies (Figure 4A).

Under normal conditions, the ectopic expression of *CqKCS2B.1* clearly affected root elongation and lateral root formation (Figure 4B). Compared with the wild-type (WT), OE#5 and OE#6 showed 21% and 12% greater root length, respectively (Figure 4C), and formed significantly more lateral roots. (Figure 4D). However, the root hairs’ density was unaffected (Appendix A). These results indicate that *CqKCS2B.1* is involved in the elongation of the primary root and lateral root development.

To explore why lateral root development was enhanced in the *CqKCS2B.1* OE lines, we detected the transcript levels of genes that are known to be involved in the regulation of lateral root development by RT-qPCR. Total RNA was extracted from the roots of 10-day-old seedlings of WT and OE#5 and OE#6. The transcript levels of genes associated with auxins, such as *GH3.3* and *AUX1*, were significantly elevated in the *CqKCS2B.1* OE lines compared with WT (Figure 5). In *Arabidopsis**, GH3.3* has been reported to encode an IAA-amido synthase that quickly responds to auxin and is positively correlated with adventitious root development [45]. *AUX1* encodes an auxin influx transporter that is involved in lateral root development [46]. The enhanced transcript levels of *GH3.3* and *AUX1* in *CqKCS2B.1* OE#5 and OE#6 seedlings suggest that *CqKCS2B.1* regulates lateral root formation by affecting auxin metabolism and transport. 

### 2.5. CqKCSB.1 Overexpression Alters Composition of Suberin Monomers in Arabidopsis Roots

Arabidopsis *KCS2* and *KCS20* are involved in the synthesis of the VLCFAs that are precursors of suberin monomers. To explore the activity of *CqKCS2B.1*, the chemical composition of suberin monomers in the roots of WT, OE#5, and OE#6 plants was analyzed by gas chromatography with flame ionization detection (GC-FID). As shown in Figure 6A, the total amounts of aliphatic suberin monomers in *CqKCS2B.1* OE#5 and OE#6 were increased by 14.48% and 15.49%, respectively, compared with that in WT. This was mainly because of a general increase in the contents of all monomers, especially the C22–C24 FA, C18:1 ω-OH acid, and C18:1 dioic acid (Figure 6C). Further analyses of the roots showed an increase of up to 9% in C20, 16% in C22, and more than 50% in C24 in the OE lines compared with WT Col-0 (Figure 6B). These results indicate that *CqKCS2B.1* is involved in the synthesis of C22–C24, the major precursors for suberin monomer production. 

### 2.6. Ectopic Expression of CqKCS2B.1 Increases the Salt Stress Tolerance of Arabidopsis 

As described above, *CqKCS2B.1* expression was quickly induced under salt stress. Therefore, the salt tolerance of *CqKCS2B.1* transgenic lines (OE#5 and OE#6) was determined by comparing their seeds’ germination rate and root growth with that of WT under salt stress conditions. For their seed germination assay, seeds of *CqKCS2B.1* OE#5, OE#6, and WT were sown on ½ MS plates supplemented with NaCl at a range of concentrations (0 mM, 50 mM, 100 mM, and 150 mM), and the germination rate was determined after 3 and 5 days. Almost all the seeds of both OE#5 and OE#6 transgenic lines and WT germinated on ½ MS under normal conditions. Compared with the germination percentage of WT seeds, i.e., 43% after 3 days and 72% after 5 days, those of the *CqKCS2B.1* OE lines were higher in all the salt treatments, even at 150 mM NaCl, i.e., 70% after 3 days and 89% after 5 days (Figure 7A–C). To investigate the survival rate of *CqKCS2B.1*, the 7-day-old seedlings were transferred to ½ MS plates containing 200 mM NaCl. The survival rates and chlorophyll contents were significantly higher in OE#5 and OE#6 than in WT (Col-0) (Figure 8a and Appendix A).

Next, the physiological status of *CqKCS2B.1* OE#5, OE#6, and WT plants were investigated under normal and salt-stressed conditions. All the lines were grown for 2 weeks under normal conditions and then treated with 150 mM NaCl for 24 h. The malondialdehyde (MDA) content reached 39.74 nmol/mg in WT under salt treatment, about double the levels in the *CqKCS2B.1* transgenic lines OE#5 and OE#6 (20.7 nmol/mg and 22.01 nmol/mg, respectively) (Figure 8B). Thus salt stress resulted in increased MDA content in WT plants compared with OE plants. These findings show that WT is more vulnerable to oxidative damage caused by salt stress.

Next, the activity of POD, which detoxifies reactive oxygen species (ROS), was determined as an index of the enzymatic antioxidant capacity of transgenic lines and WT. The POD activity was sufficiently higher in the *CqKCS2B.1* OE#5 and OE#6 lines than in WT (Figure 8C), indicating that overexpression of *CqKCS2B.1* resulted in increased POD activity and lower ROS levels under salt stress. Proline scavenges ROS produced under salt stress, allowing the plant to recover more quickly. The proline level was higher in the *CqKCS2B.1* OE transgenic plants than in WT under control conditions, and salt stress increased the proline content by 2.2-fold in OE#5 and 1.9-fold in OE#6 (Figure 8D). Together, these results showed that overexpression of *CqKCS2B.1* in *Arabidopsis* results in increased POD activity and proline content, and decreased MDA content under salt stress, and increased salt tolerance.

## 3. Discussion

In plants, VLCFAs are synthesized by the FAE complex consisting of KCS, KCR, *PAS2*, and *CER10*. The KCS enzyme is the first and rate-limiting step in VLCFA biosynthesis and its substrate is elongated to form VLCFAs [47]. In this study, we identified 33 *CqKCS* gene-encoding FAE-type enzymes in quinoa (Appendix A). The phylogenetic analyses revealed two homologs of *AtKCS2/AtKCS20* in quinoa, *CqKCS2A.1* and *CqKCS2B.1*. A previous study showed *Arabidopsis KCS2* and *KCS20* catalyzed the formation of C22–C24 VLCFAs when heterologously expressed in *Saccharomyces cerevisiae* [48]. Subsequent analysis of *Arabidopsis* single and double mutants of *KCS2* and *KCS20* revealed that both genes are functionally redundant in cuticular wax biosynthesis and the production of C22–C24 VLCFA derivatives in suberin monomers [15]. The overexpression of *KCS2* homologs in *Arabidopsis* resulted in increased VLCFA production and the significant accumulation of cuticular wax [49]. Additionally, molecular characterization of *Citrus sinensis KCS20* by ectopic expression in *Arabidopsis* and yeast indicated that it catalyzes the synthesis of C22 and C24 VLCFAs, similar to *KCS2* or *KCS20* in *Arabidopsis* [50]. In our study, overexpression of *CqKCS2B.1* in *Arabidopsis* also led to increased contents of C22 and C24 VLCFAs and their derivatives in suberin monomers, suggesting that the function of *KCS2* homologs is evolutionary and functionally conserved among diverse plant species. 

There is a large body of evidence showing that VLCFA or its derivatives act as bioactive signals to mediate various developmental processes [51,52]. For example, VLCFAs or their derivatives regulate signals that determine the ability of the pericycle to form callus, and therefore determine the plant organ’s regeneration capacity [36]. Auxin is one of the key phytohormones that direct the formation of lateral roots from the pericycles [53,54]. Previous studies have shown that VLCFAs are required for polar auxin transport during plant development and tissue patterning [55], and that MPK14-mediated auxin signaling influences lateral root growth in Arabidopsis, which is mediated through ERF13-regulated VLCFA synthesis [56]. In our study, we found that the number of lateral roots was significantly increased and the transcript levels of auxin-associated genes such as *AUX1* and *GH3.3* were dramatically increased in the *Arabidopsis CqKCS2B.1* OE transgenic lines. These results indicate that *CqKCS2B.1* increases the contents of VLCFAs, which may function as signaling molecules to regulate the expression of auxin-associated genes, increase the metabolism and transport of auxin from the source to sink, and increase lateral roots’ synthesis. Additionally, various studies have shown that VLCFAs regulate plant growth and development processes, such as cell elongation, through an ethylene-mediated signal pathway. For example, overexpression of *GhKCS13* in *Arabidopsis* was found to promote stem cell elongation by regulating the transcript levels of *ACO* (encoding an enzyme catalyzing ethylene biosynthesis) [57]. Therefore, the increased primary root’s length in the *CqKCS2B.1* OE lines might be caused by the activation of the ethylene signaling pathway by the increased VLCFAs.

The VLCFAs also regulate tolerance to abiotic stresses in plants. For example, overexpression of KCS homologs from several species was found to increase VLCFAs’ content in plants and enhance tolerance to drought or salt stress [9,58]. Similar results were observed in our study. That is overexpression of *CqKCS2B.1* visibly improved the salt tolerance of *Arabidopsis* as indicated by increased seed germination and survival rates. In addition, compared with WT, the Arabidopsis *CqKCS2B.1* OE lines showed significantly increased POD activity and proline levels and lower MDA contents under salt stress. Previously, VLCFAs were reported to mediate auxin distribution and regulate the auxin signal, thereby mediating various physiological processes during adaption to salt stress [56,59]. In our study, we found that the transcript levels of the auxin-associated genes *AUX1* and *GH3.3* were significantly increased in *Arabidopsis CqKCS2B.1* OE transgenic lines, suggesting that *CqKCS2B.1* improves tolerance to salt stress in *Arabidopsis* by activating the auxin signal. Additionally, overexpression of *CqKCS2B.1* in *Arabidopsis* significantly increased the contents of aliphatic suberin monomers in roots. Previous studies have shown that these monomers play a role in protecting plants against excessive apoplast movement of water and solutes into the root stele [60]. The increased tolerance might also be due to the increase in the suberin layer, which reduces the apoplast movement of Na^+^ ions, with the results reported by Lokesh et al. [61].

In plants, seed coats provide structural support to protect the embryo and other parts of the seed from biotic and abiotic environmental stresses [62]. Among the five cell layers in the seed coat, the lipophilic suberin layer contains primary sealing compounds that make the seed coats impermeable to water and nutrients. Previously, it was reported that changes in the suberin layer of the seed coat affect the seed’s viability and germination rate. For example, compared with WT, a *gpat5* mutant that altered the aliphatic composition of suberin in the seed coat showed a lower germination rate under salt stress [63]. Similarly, the seed coat permeability of the *far1 far4 far5* mutant was significantly increased, which provides further evidence for the role of suberin as a primary barrier in the seed coat [64]. Compared with seed coats that lack suberin, those containing suberin have higher contents of C24 VLCFAs and their derivatives [65,66]. In the present study, the germination rates of the *CqKCS2B.1* OE were higher than those of WT under salt stress, probably because of the increased accumulation of VLCFA in the suberin layer of the seed coat.

## 4. Materials and Methods

### 4.1. Identification of CqKCS Genes

The conserved domains of the AtKCS family were obtained from the PFAM protein families database (http://pfam.xfam.org/, accessed on 15 January 2021). Then, BLASTp was used to retrieve the standalone similarity sequence, and the conserved protein domain was entered into the Chenopodium database ChenopodiumDB (https://www.cbrc.kaust.edu.sa/chenopodiumdb/index.html, accessed on 15 January 2021). The genomic and proteomic sequences of *C. quinoa* were retrieved. The sequences of *Arabidopsis* KCS proteins were obtained from The *Arabidopsis* Information Resource (TAIR) database (https://www.arabidopsis.org/, accessed on 15 January 2021). A hidden Markov models (HMM) EMBL-EBI PFAM search was carried out for domain detection, https://www.ebi.ac.uk/Tools/hmmer/, accessed on 15 January 2021).

The conserved domain of KCS from quinoa was identified using tool at the Conserved Domain Database (CDD) [67] of the National Center for Biotechnological Information (NCBI) (https://www.ncbi.nlm.nih.gov/Structure/cdd/wrpsb.cgi, accessed on 15 January 2021) [68], whereas, the protein domains were displayed using TBtool [69]. Various other physicochemical properties of the 33 putative CqKCS proteins were investigated using Expasy ProtParam (https://www.expasy.org/, accessed on 15 January 2021). 

### 4.2. Phylogenetic Analysis 

The KCS sequences identified from quinoa (*C. quinoa*) and *A. thaliana* were aligned using Clustal X in MSA (Multiple Sequence Alignment Version 2.0) [70]. A preliminary phylogenetic tree was constructed using MEGA 11 (Molecular Evolutionary Genetic Analysis version 11) with the neighbor-joining method [71]. The bootstrap analysis was performed with 1000 replicates, the Poisson model, and pairwise deletion to statistically assess each node. The tree was visualized using Interactive Tree Of Life (itol) (https://itol.embl.de/, accessed on 23 March 2022) [72]. 

### 4.3. Gene Duplication Analysis

The evolutionary rates, Ka (non-synonymous substitution rate) and Ks (synonymous substitution rate), were estimated using the Ka/Ks calculation tool (http://services.cbu.uib.no/tools/kaks, accessed on 23 March 2022). Duplicated gene pairs were illustrated in Circos using TBtool.

### 4.4. Plant Materials and Growth Conditions

The white quinoa seeds Q75 (CM639) were acquired from Shandong Provincial Key Laboratory of Plant Stress, College of Life Sciences, Shandong Normal University, Jinan 250014, China. The quinoa seeds were surface sterilized by NaClO and stratified on Murashige and Skoog (½ MS) and grown for 3 days, and for tissues analysis, the plant was transferred to soil. However, the Arabidopsis seeds were surface sterilized and germinated on ½ MS for 10 days and then grown on the soil at 24 °C day/22 °C night, 16 h light/8 h dark, whereas for phenotypic analyses of transgenic *CqKCS2B.1* lines in comparison to wild-type (Col-0), the seeds were stratified on ½ MS plates. The root length and number of lateral roots were investigated after 10 days. The root length and number of lateral roots were measured using ImageJ software [73].

The tobacco (*Nicotiana benthamiana*) seeds were sown in soil and were grown at 25 °C for 16 h day/8 h night. The plants were grown until the leaves were expanded enough (normally 6 leaves) and suitable for injection.

### 4.5. RNA Extraction, RT-PCR and RT-qPCR Analysis

The *CqKCS2* relative expression among different tissues of the quinoa plant was studied. Total RNA was extracted from 1-month-old quinoa plant roots, stems, and young leaves, as well as from mature leaves and inflorescences of quinoa using RNA Easy Fast plant Tissue kit (TIANGEN BIOTECH, Beijing, China), and the cDNAs were synthesized using Superscript™ III Reverse Transcriptase kit (Invitrogen, Waltham, MA, USA), whereas, for the lateral roots, associated auxin genes’ relative expression of 10-day-old transgenic *CqKCS2B.1* OE lines and WT (Col-0) seedlings’ root tissues were used to carry out RT-qPCR. The total RNA was extracted and cDNA was synthesized. RT-qPCR analysis was conducted separately for the different tissues of the quinoa plant and the lateral roots associated with Arabidopsis genes. The RT-qPCR was performed using a 2× QuantiTect SYBR Green PCR mix (TIANGEN BIOTECH, Beijing, China) and light cycler^®^ 96 SW 1.1 real-time PCR system (Roche, Basel, Switzerland). The relative gene transcript level was measured as 2^−∆∆*Ct*^ [74].

However, the expression of *CqKCS2A.1* (*AUR62006329-RA*) and *CqKCS2B.1* (*AUR62026367-RA*) under 300 mM salt stress was analyzed by semi RT-PCR. The gene-specific primers were synthesized for *CqTUB6* (*AUR62018378-RA*) and used as a reference gene, whereas, *CqCMO* (*AUR62043230-RA*) was used as a salt stress response gene [44]. All the primers used in this study are listed in (Appendix A).

### 4.6. Subcellular Localization of CqKCS2B.1

The *Pro35S::CqKCS2B.1-YFP* and *Pro35S::CqKCS2B.1-YFP* were constructed to determine the subcellular localization of *CqKCS2B.1* and *CqKCS2B.1* proteins, respectively. The full-length coding regions of CqKCS2A.1 (AUR62006329-RA) and *CqKCS2B.1* (*AUR62026367-RA*) were amplified from the cDNA of quinoa roots and then induced into the *pFGC-eYFP* vector digested with BamHI to C-terminally fused YFP using a 2x Seamless Cloning Mix (Biomed, Beijing, China). The positive clones were identified and recombinant plasmids were sequenced to avoid the error from base pair mismatch in the PCR step. The construct was co-expressed with the endoplasmic reticulum localized marker CD3-959 (ER-mCherry) [75] in *Nicotiana benthamiana* leaves [76]. The signals of YFP and mCherry were obtained using confocal laser scanning microscopy (Leica, TCS SP8 MP) with the excitation/emission wavelengths: YFP (514/535–580 nm) and mCherry (561/590–620 nm).

### 4.7. Generation of CqKCS2B.1 Overexpression Lines in Arabidopsis

The above-mentioned *Pro35S::CqKCS2B.1-YFP* construct was introduced into Arabidopsis wild-type (Col-0) by *Agrobacterium tumefaciens* GV3101-mediated transformation [77]. At least 13 positive T1 transgenic lines were obtained from the transformed seeds under the screening of the herbicide Basta. Subsequently, total protein was extracted from seedlings of transgenic *CqKCS2B.1* OE lines by using the immunoprecipitation buffer containing 100 mM Tris-HCl (pH 7.5), 75 mM NaCl, 1 mM EDTA, 0.1% Triton X-100, and 1% protease inhibitor cocktails at 4 °C. The amounts of total protein were then quantified via Bradford’s assay before Western blotting (anti-GFP) [78,79,80]. According to the relatively higher expression level of *CqKCS2B.1*-YFP protein, two lines (OE#5 and OE#6) were selected and T2 homozygous generation was obtained for further investigation.

### 4.8. Determination of Salt Tolerance of WT and CqKCS2B.1 OE Lines

We determined the seed germination rate, root growth, and number of lateral roots in *CqKCS2B.1* OE transgenic plants and WT under salt stress. For the germination assay of *CqKCS2B.1* transgenic lines OE#5, OE#6, and WT (Col-0), 80 seeds per plate were sown on ½ MS medium supplemented with varying concentrations of NaCl (0, 50 mM, 100 mM, and 150 mM). The seeds were incubated for 5 days in total, and the germination rate was recorded after 3 days and 5 days in the light. The resistance of *CqKCS2B.1* transgenic OE lines and WT to dehydration under salt stress was determined. For these analyses, 1-week-old seedlings were transferred to ½ MS plates supplemented with 200 mM NaCl. 

For analyses of other parameters in *CqKCS2B.1* OE and WT plants under salt stress, plants of *CqKCS2B.1* OE#5, OE#6, and WT (Col-0) were grown for 2 weeks on ½ MS plates and then treated with 0 and 150 mM NaCl for 24 h. The MDA content, POD activity, and proline content were determined using corresponding detection kits (Nanjing Jiancheng Bioengineering Institute, Nanjing, China) according to the manufacturer’s instructions. All data shown are the mean of three repeated experiments. In addition, 14-day-old WT and CqKCS2B.1 transgenic lines were used to analyze the total chlorophyll content under normal conditions. A total of 0.2 g of fresh plant tissue was mixed with 80% acetone, and the absorption was measured using a spectrophotometer at 663.2 and 646.8 nm [81]. Differences between OE lines and WT were considered significant at *p* < 0.05.
Chlorophyll “a” (g/mL) = 12.25 × A_663.2_ − 2.79 × A_646.8_
Chlorophyll “b” (g/mL) = 21.50 × A_646.8_ − 5.1 × A_663.2_
Total Chlorophyll (mL) = 7.15 × A_663.2_ + 18.71 × A_646.8_

### 4.9. Suberin Content Analysis in Plant Root

The root suberin content of a four-week-old plant was determined following the instructions of the Jenkin and Molina [82] procedure. The plants were carefully removed from the soil, mechanical damages were avoided by removing shoots from the roots, and then carefully washed. The suberin aliphatic content was analyzed using the four steps followed by data analysis. In the first step, tissue dilapidation was performed in which the solvent-soluble lipids were removed. The second step was depolymerization wherein the tissue was depolymerized into its component lipids’ monomers by the sodium methoxide catalysis methanolysis. The solvent fraction was washed with saline, the aqueous phase was removed, Na_2_SO_4_ was added, and samples were transferred to the small glass tube. The third step comprises of preparation of derivatives and transferring them for GC analysis.

### 4.10. Statistical Analysis

Three replications were conducted for each experiment in a completely randomized design. Data processing and chart drawing were performed using GraphPad prism 8. Data were analyzed using ANOVA and the student’s *t*-test. Significant differences were indicated by *p* < 0.05. GraphPad Prism 8 was used for statistical computations.

## 5. Conclusions

We identified KCS-encoding genes from *C. quinoa.* of these genes, the two homologs of *AtKCS2,* namely *CqKCS2A*.1 and *CqKCS2B*.1, showed increased expression in quinoa seedlings under salt stress. Transgenic *Arabidopsis* ectopically expressing the ER-localized *CqKCS2B*.1 showed significantly increased synthesis of VLCFAs and their derivatives in roots. The upregulation of auxin-associated genes including *GH3*.3 and *AUX1* in the transgenic lines was related to increased root length and more lateral roots. Together, these findings indicate that *CqKCS2B*.1 regulates VLCFAs’ production to increase lateral root development and enhance tolerance to salt stress.

## Figures and Tables

**Figure 1 ijms-23-13204-f001:**
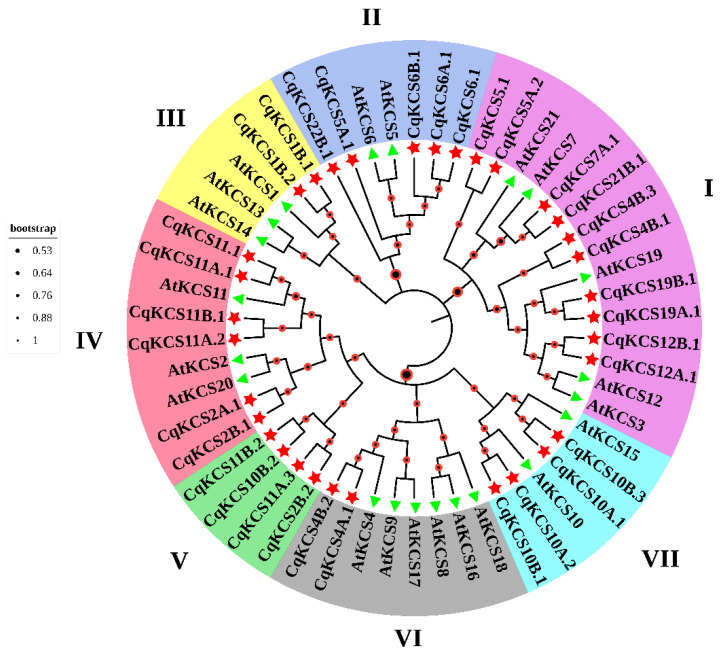
The evolution and diversification of the *C. quinoa* KCS gene family. The neighbor-joining method was used in MEGA11 to construct an unrooted phylogenetic tree of KCS (FAE1-type) using Arabidopsis and quinoa protein sequences. Bootstrap values are represented by red/black circles. Different colors in the outer ring indicate different groups.

**Figure 2 ijms-23-13204-f002:**
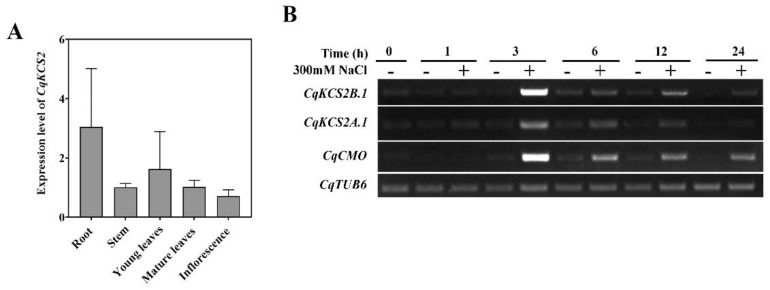
Transcript profile of *CqKCS2A.1* and *CqKCS2B.1*. (**A**) Detection of total *CqKCS2* transcript levels in different tissues of quinoa: 1-month-old plant root, stem, leaves, mature plant leaves, and inflorescences. (**B**) Semi-quantitative RT-PCR analyses of *CqKCS2A.1* and *CqKCS2B.1*. Total RNA was extracted from 3-day-old quinoa seedlings grown in vitro and exposed to 300 mM NaCl for 0, 1, 3, 6, 12, and 24 h. In each case 1 µg total RNA was used as a template.

**Figure 3 ijms-23-13204-f003:**
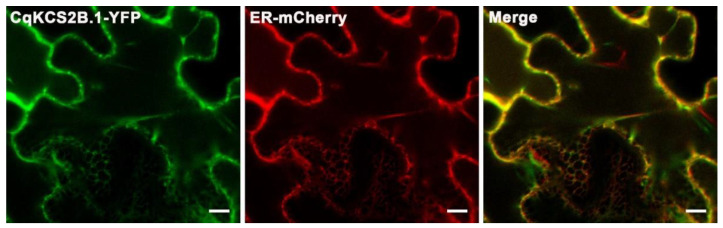
Localization of *CqKCS2B.1*-YFP in the endoplasmic reticulum. Signals of *CqKCS2B.1*-YFP and ER-mCherry were merged in the epidermal cells of *Nicotiana* leaves. Bar = 10 µm.

**Figure 4 ijms-23-13204-f004:**
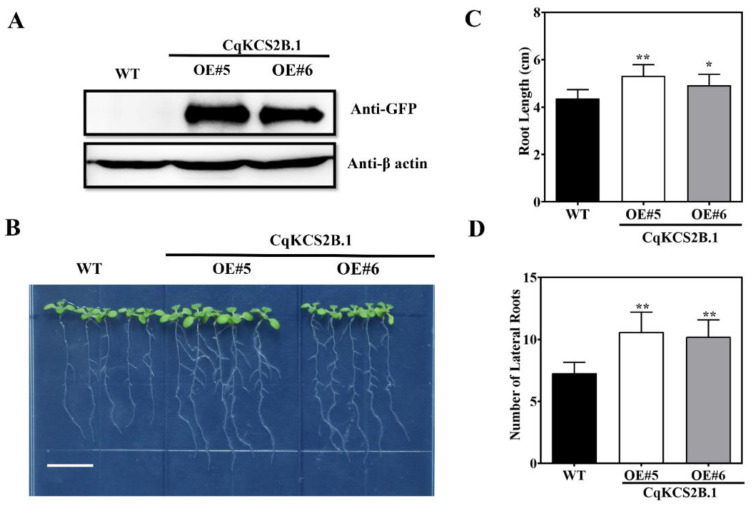
Ectopic expression of *CqKCS2B.1* increases root elongation and lateral root synthesis. (**A**) Protein profiling of CqKCS2B.1 overexpressing lines OE#5 and OE#6, and wild-type (WT). Blot was probed with anti-β Actin and anti-GFP. (**B**) Root length and the number of lateral roots in WT and *CqKCS2B.1* OE lines. The image shows OE lines (OE#5 and OE#6) seedlings on vertically oriented plate of ½ MS agar medium. (**C**,**D**) Root length and the number of lateral roots of 7-day-old seedlings of WT and *CqKCS2B.1* OE lines (OE#5 and OE#6) at 3 d after transfer to a new ½ MS plate. Values are mean ± SD of 15 independent measurements. Significant differences were detected by ANOVA. * and ** indicate a significant difference between different transgenic line and WT at *p* < 0.05. and *p* < 0.01, respectively. Scale = 1 cm.

**Figure 5 ijms-23-13204-f005:**
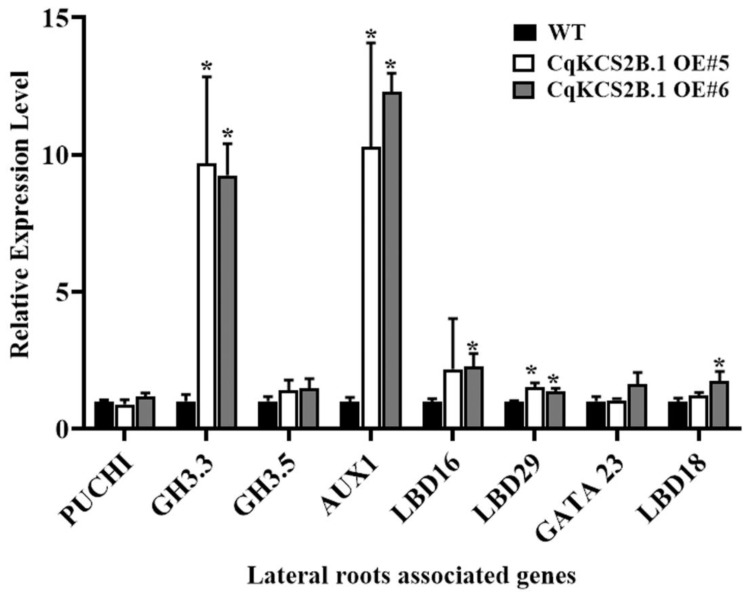
RT-qPCR analysis of lateral root-associated genes. Analyses were conducted using RNA extracted from the roots of 10-day-old transgenic plants of CqKCS2B.1 OE#5 and OE#6, and WT (Col-0). Gene transcript levels were normalized to that of *AtACTIN 2* transcript. Values are the mean ± SD of three repeats. * indicates significant differences (*p* < 0.05) between transgenic lines and WT (Student’s *t*-test).

**Figure 6 ijms-23-13204-f006:**
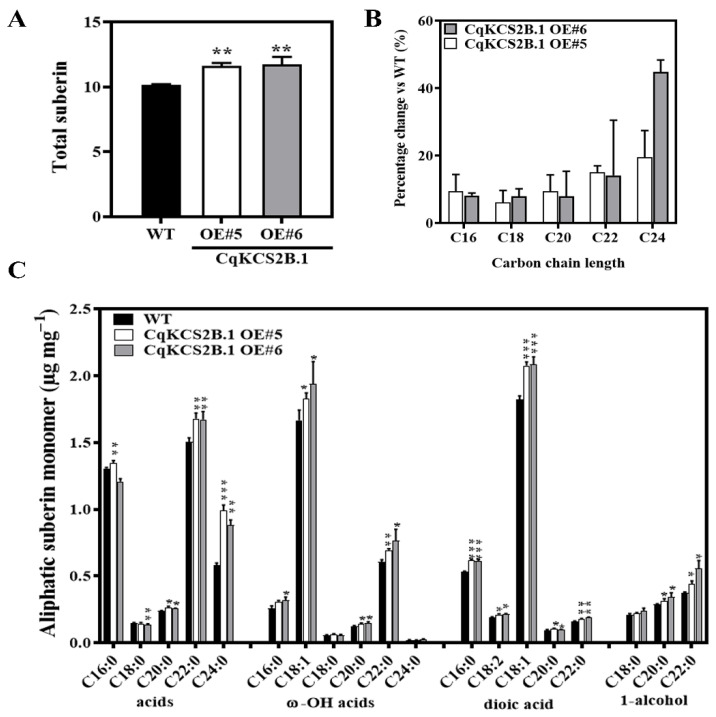
Gas chromatographic analysis of aliphatic suberin monomers and endodermal suberin in roots of transgenic Arabidopsis and WT (Col-0). (**A**) Total suberin content in roots of 4-week-old CqKCS2B.1 OE#5, OE#6, and WT plants. (**B**) Distribution of aliphatic monomers of suberin according to chain length. (**C**) GC-MS results grouping fatty acids in WT and OE lines according to chain length, saturation status, and presence of ω-OH acid, dioic acid, and 1-alcohol additions. Values are mean ± SD. *, ** and *** indicate significant differences between OE lines and WT at *p* < 0.05, *p* < 0.01, and *p* < 0.001 respectively (Student’s *t*-test).

**Figure 7 ijms-23-13204-f007:**
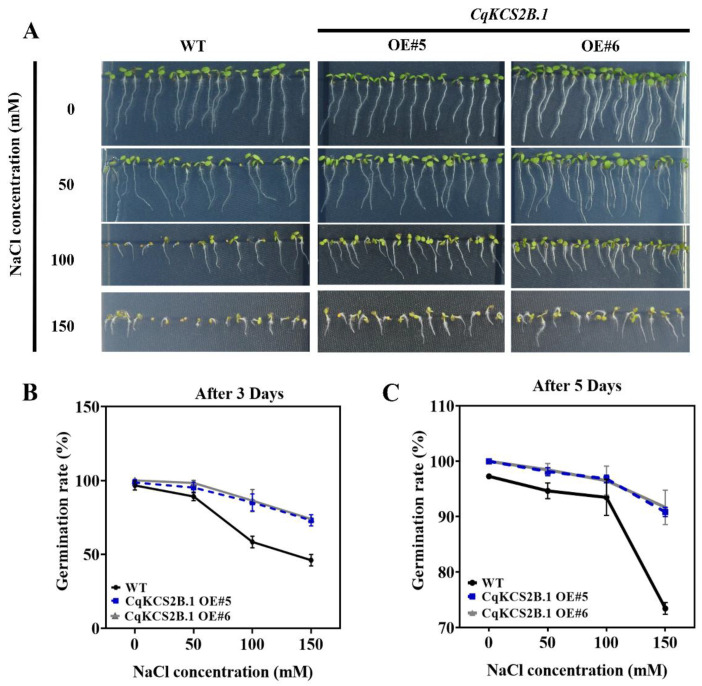
Seeds’ germination rates of WT and *CqKCS2B.1* OE lines under varying degrees of salt stress. (**A**) Germinating seeds of WT (Col-0), *CqKCS2B.1* OE#5, and OE#6 at 5 d after exposure to light after treatment with NaCl at t 0 to 150 mM. (**B**) Seed germination percentage on medium containing 0, 50, 100, and 150 mM NaCl treatment after 3 d in the light. (**C**) Seed germination percentage on medium containing 0, 50, 100, and 150 mM NaCl at 5^th^ day in the light. Each value represents the mean ± SD of 50 independent measurements. Statistical significance was evaluated by ANOVA.

**Figure 8 ijms-23-13204-f008:**
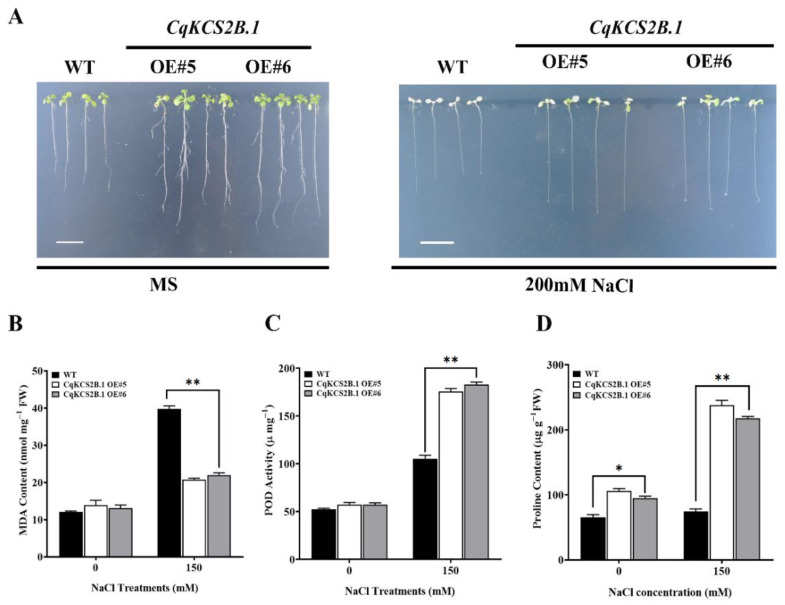
Phenotypic analyses of *CqKCS2B.1* OE transgenic *Arabidopsis* seedlings under salt stress. (**A**) 1-week-old seedlings of WT, OE#5, and OE#6 were treated with 200 mM NaCl until the WT seedlings were completely dehydrated and then the plants were photographed. Two-week-old plants were treated with 150 mM NaCl for 24 h before determining (**B**) MDA content, (**C**) POD activity, and (**D**) proline content. Values are mean ± SD of 20 independent measurements. Statistically significant differences were detected by ANOVA. * and ** indicate significant differences between transgenic lines and WT at *p* < 0.05 and *p* < 0.01. Scale = 1 cm.

## Data Availability

Data are available from the author on request.

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
