# Peer review of "Overexpression of *β-Ketoacyl CoA Synthase 2B.1* from *Chenopodium quinoa* Promotes Suberin Monomers’ Production and Salt Tolerance in *Arabidopsis thaliana"

_ijms, 2022, doi:10.3390/ijms232113204_

Round 1
Reviewer 1 Report
It is an interesting topic for practice. However, the following comments have to be addressed for author’s consideration. I suggest minor revisions to your manuscript before re-considering publication in the attached file herein. The English quality must be improved. The statistical analyses should be wrote in methods, revised along with all figures.

Author Response
General Comments:
Major improvement in the English grammar issues
Author’s response. The grammar and other issues related to language were corrected throughout the manuscript with the help of a native speaker and also according to the reviewer's comments.
Captions need more detail including abbreviations. Spacing and statistical indicator changes.
Author’s response. The capitalization, spacing and abbreviation of all the abbreviated text within the manuscript have been modified according to the reviewer's suggestions.
Line 30: the key words should be check and extra keywords to be incorporated, and it’s recommended to don’t use abbreviations e.g., (Salt tolerance, Suberin monomer, β-Ketoacyl CoA Synthase 2B, Very-long-chain fatty acids, Lateral root, Chenopodium quinoa).
Author’s response. The keywords have been checked and additional keywords were added as follows:
Keywords: Chenopodium quinoa; β-Ketoacyl CoA Synthase2B.1; Very-Long-Chain Fatty Acids; Suberin monomers; Lateral root, Salt tolerance.
Line 91-98: I think the aims un clear, therefore, state the objectives more precisely and according to the title, also match it with conclusions.
Author’s response. The aims of this study were revised and included in lines 93-98 of the revised manuscript and the conclusion was also modified and matched to the aim accordingly.
Line 196-198: un clear! Also, I noticed there are differences in the expression for the statistical analyses for different figures in the manuscript, please revise. Also, add the mean and SD, along with this expression.
Author’s response. The differences in the expression for the statistical analyses for different figures in the manuscript were changed overall. The mean and SD were also added with all expressions.
Line 285: the ** indicates the P<0.05 What about *?? You should write all symbols in the figures, please solve this problem in all figures for the statistical analyses.
Author’s response. The differences in the expression for the statistical analyses for different figures in the manuscript were changed overall. The mean and SD were also added with all expressions. The ** was mistakenly written as p<0.05, however it was p<0.01, and * was p<0.05. all the symbols of the figure were written in the captions in revision.
Special comments
Write the statistical analyses in the material and methods.
Authors response. Thanks for your comment. The statistical analysis section has been added in the materials and methods section (Line 487-491 in the revised manuscript).
Overall: the conclusions should be revise and editing for spelling mistakes, specify the achieved aims and conclude the future perspectives.
Author response. Thanks for your comment. The aims were mentioned as it was pointed out by the reviewer. The conclusion has been overall revised and spelling mistakes were omitted. The achieved aims and future perspectives were also interpreted in the conclusion section.
Check the references and uniform the format according to the journal format, since sometimes you abbreviate the journal name, while other times no.
Author response. All the references were formatted uniformly in the reference style of IJMS (MDPI) and all the journal names in all the references were changed to the abbreviated form.
The English quality must be improved.
Author response. We agree. The English grammar and other language-related issues were improved thoroughly in the manuscript with the help of a native speaker.
Reviewer 2 Report
It is a nice study working with a traditional non-model crop. The authors show data for a compressive functional characterization of quinoa genes involving VLCFA biosynthesis. Different evidence lines, including expression level quantification, physiological responses to stress tests, and metabolite analysis, support the discussion and conclusions. Following, I enlist my issues.
Lines 140-156; 407-408. Were the primers used for CqKCS2A.1 and CqKCSB.1 specific? Why was it possible to use them for determining expression level to abiotic stress (Figure 2B) but not for tissue-specific expression? As indicated in the M&M section (407-408), the same pair of primers were used in both cases. I suggest to carry out these experiments using specific primers harbored in the UTR sequences. Figure 2 caption: CK = mock or control? CK is often used as abbreviation (abb) for cytokinins. For no confusion, please use another abb for the control treatments.
Lines 173-174. CqKCS subcellular localization. What is the subcellular localization of CqKCS2A.1 protein? We may hypothesize it also a ER-locate protein but the evidence is missing. Do you make any construct to address this point? Why for this analysis you selected only to CqKCS2B.1
Figure 4. Considering VLCFA function in plant epidermis (line 42), did you look at root hair in OE lines? In Figure 4B, making a zoom-in, it seems OE line roots have more root hairs. During NaCl stress, the line OE6 seems to have a higher density of root hairs than WT (Figure 7A).
Line 206. Quick auxin-responsive? How? Explain this phrase. Any time-course analysis involving auxin hormone was carried out.
Lines 261-262. Please, check the spelling of this phrase. Something is missing after i.e.
Discussion. How about the biological roles of VLCFA in development regulation, specifically cell elongation? Your data show a key function of CqKCS2B.1 in root development affecting the length of primary roots under control conditions
(Figure 4B, 4C) and salt stress (Figure 8A).
Lines 134-137. Figure 1. Bootstrap values are not indicated. Blue circles?
Line 143. Please, remove that comma after quinoa plant.
Line 161. Please, split the words in vitro.
Line 174. Please, split words 10uM.
Line 185. Please, split word Figure 4C.
Line 196, 279. No Capital letter for Statistical
Lines 190-198. Figure 4B, please indicate the scale value for the bar.
Line 216, 238. Student's T-test not students t-test.
Line 232. Figure 6C. Please, use superscript for 1 in the legend of the vertical axis.
Line 232. No Capital letter for Chromatographic.
Line 261. peroxidase (POD), instead POD (Peroxidase)
Line 256. Please, declare the abbreviation MDA before use.
Figure 8. Please, no Capital letters for content, activity in Figure 8B, 8C, and 8D.
Line 302. No Capital letter for overexpressing.
Author Response
Lines 140-156; 407-408. Were the primers used for CqKCS2A.1 and CqKCSB.1 specific? Why was it possible to use them for determining expression level to abiotic stress (Figure 2B) but not for tissue-specific expression? As indicated in the M&M section (407-408), the same pair of primers were used in both cases. I suggest to carry out these experiments using specific primers harbored in the UTR sequences. Figure 2 caption: CK = mock or control? CK is often used as abbreviation (abb) for cytokinins. For no confusion, please use another abb for the control treatments.
Author response. Thanks for your comment. We tried to design primers in the UTR regions of both genes. However, the CqKCS2A.1 and CqKCS2B.1 have a high percentage of homologous sequences (95.69%), as shown in revised Supplementary Figure 3. So, in this case, the detection of a total of two genes expression was carried out using qRT-PCR.
The semi-RT-PCR primers for CqKCS2A.1 and CqKCS2B.1 were confirmed to be specific by using vectors of Pro35S::CqKCS2A.1-YFP and Pro35S::CqKCS2B.1-YFP as templates (shown in revised Supplementary Figure 4). Additionally, we also performed the semi-RT-PCR to detect expression levels of both genes to abiotic stress (shown in revised Supplementary Figure 5), which is consistent with the RT-qPCR results.
Lines 173-174. CqKCS subcellular localization. What is the subcellular localization of CqKCS2A.1 protein? We may hypothesize it is also an ER-locate protein but the evidence is missing. Do you make any construct to address this point? Why for this analysis you selected only to CqKCS2B.1
Author response. CqKCS2A.1 and CqKCS2B.1 were found to be 95.69% homologs. So, it can be hypothesized that the CqKCS2A.1 is also localized to the ER. We did make the Pro35S::CqKCS2A.1-YFP construct and found that CqKCS2A.1 is localized to the ER by transient expression in tobacco leaves (shown in revised Supplementary Figure 6).
After the expression analysis of both genes under salt stress, we found that CqKCS2B.1 show a higher expression level than CqKCS2A.1 in response to salt stress. Therefore, we chose the CqKCS2B.1 for further functional analysis.
Figure 4. Considering VLCFA function in plant epidermis (line 42), did you look at root hair in OE lines? In Figure 4B, making a zoom-in, it seems OE line roots have more root hairs. During NaCl stress, the line OE6 seems to have a higher density of root hairs than WT (Figure 7A).
Author response. The VLCFA function in plant epidermis has been reported previously, such as in the case of GhKCS13/CER6 encoding 3-Ketoacyl CoA synthase, which is involved in maximizing the extensibility of cotton fiber and also various cell types in Arabidopsis [1]. CsKCS6 ectopic expression in Arabidopsis increases the number of trichomes in the stem and leaves [2]. From the previous studies, we can conclude that KCS or VLCFA-associated genes are involved in regulating epidermal cell structures such as cotton fibers and trichomes. However, VLCFA regulating the root hairs is largely unknown.
The experiment was carried out to analyze the root hairs of CqKCS2B.1 transgenic lines and WT under normal and salt stress conditions (100 mM NaCl) after one week. However, the results showed there was no significant difference between WT and the transgenic line of CqKCS2B.1 (shown in revised Supplementary Figure 7).
Line 206. Quick auxin-responsive? How? Explain this phrase. Any time-course analysis involving auxin hormone was carried out
Authors response. Thank you for highlighting this point. The phrase was incorrect and has been rephrased as follows:
Line 219-221
In Arabidopsis, GH3.3 is previously reported to encode an IAA-amido synthase that quickly responds to auxin and is positively correlated with adventitious root development.
Discussion. How about the biological roles of VLCFA in development regulation, specifically cell elongation? Your data show a key function of CqKCS2B.1 in root development affecting the length of primary roots under control conditions (Figure 4B, 4C) and salt stress (Figure 8A).
Author response. The VLCFA influence multiple hormone signal pathways to participate in plant development regulation including cell proliferation, differentiation and elongation. The corresponding content is associated with the longer primary roots in CqKCS2B.1 OE lines was discussed as follows
Line 331 to 337:
Additionally, various studies have shown that VLCFAs regulate plant growth and development processes, such as cell elongation, through an ethylene-mediated signal pathway. For example, overexpression of GhKCS13 in Arabidopsis was found to promote stem cell elongation by regulating the transcript levels of ACO (encoding an enzyme catalyzing ethylene biosynthesis) [1]. Therefore, the increased primary roots length in the CqKCS2B.1 OE lines might be caused by the activation of the ethylene signaling pathway by the increased VLCFAs.
Lines 134-137. Figure 1. Bootstrap values are not indicated. Blue circles?
Author response. Thank you for mentioning the bootstrap values. The phylogenetic tree was constructed again with a bootstrap value chart on the left side (Figure 1 in the revised manuscript).
Capitalization, spelling mistakes, spacing, phrase corrections and other English grammar-associated issues.
Author’s response. The capitalization, spelling mistakes, spacing, and phrase corrections within the manuscript have been modified according to the reviewer's suggestions.
References
- Qin, Y.-M.; Hu, C.-Y.; Pang, Y.; Kastaniotis, A.J.; Hiltunen, J.K.; Zhu, Y.-X. Saturated Very-Long-Chain Fatty Acids Promote Cotton Fiber and Arabidopsis Cell Elongation by Activating Ethylene Biosynthesis. Plant Cell 2007, 19, 3692-3704, doi:10.1105/tpc.107.054437
- Guo, W.; Wu, Q.; Yang, L.; Hu, W.; Liu, D.; Liu, Y. Ectopic Expression of CsKCS6 From Navel Orange Promotes the Production of Very-Long-Chain Fatty Acids (VLCFAs) and Increases the Abiotic Stress Tolerance of Arabidopsis thaliana. Front Plant Sci 2020, 11, 564656, doi:10.3389/fpls.2020.564656
Reviewer 3 Report
The topic of this manuscript is up-to-date and attract wide audience of researchers and breeders. From Introduction, the significance of salt stress is missing, only two sentences describe these data. In the results, it can be seen that after 200 mM NaCl treatment Arabidopsis seedlings showed chlorotic leaves in all lines. Did authors check the photosynthesis or chlorophyll content of seedlings? And what is the explanation of this response?
Discussion and Conclusion parts are well written, however the whole manuscript needs serious English language revisions.
Author Response
In the results, it can be seen that after 200 mM NaCl treatment Arabidopsis seedlings showed chlorotic leaves in all lines. Did authors check the photosynthesis or chlorophyll content of seedlings? And what is the explanation of this response?
Author response. Thank you for raising this question. The total chlorophyll content of CqKCS2B.1 seedling was comparatively higher than that of WT seedlings (shown in revised Supplementary Figure 8). Similar results were achieved previously by Chen et al. [1] overexpression of KCS1 or AKR2A increases VLCFA and improves Fv/Fm, chlorophyll content, and provides tolerance against chilling stress. Whereas, (Fv/Fm) is the maximal photochemical efficiency of Photosystem II.
According to, Stepien and Johnson [2] Arabidopsis thaliana's total chlorophyll content decreases by up to 48% after being treated with 150 mM of salt stress. The salt stress causes degradation of chlorophyll structure by decreasing 5-aminolaevulinic acid (ALA) synthesis than chlorophyllase in Helianthus annuus [3]. Plant roots absorb salt, minerals, and water from the soil. The casparian strip and suberin lamellae are key features in roots that function as a first barrier and reduce the salt uptake from the environment into the vasculature by up to 95% and also reduce water loss or backflow [4-6]. CqKCS2B.1 transgenic lines showed increased total suberin content, which might reduce the salt uptake by the roots. However, in our experiment, we find that the chlorotic leaves were among all the seedlings types i.e., WT, and CqKCS2B.1 lines but the increased suberin can help the CqKCS2B.1 transgenic plants to survive for a longer period and maintain the chlorophyll structure.
References
- Chen, L.; Hu, W.; Mishra, N.; Wei, J.; Lu, H.; Hou, Y.; Qiu, X.; Yu, S.; Wang, C.; Zhang, H.; et al. AKR2A interacts with KCS1 to improve VLCFAs contents and chilling tolerance of Arabidopsis thaliana. Plant J. 2020, 103, 1575-1589, doi:10.1111/tpj.14848.
- Stepien, P.; Johnson, G.N. Contrasting Responses of Photosynthesis to Salt Stress in the Glycophyte Arabidopsis and the Halophyte Thellungiella: Role of the Plastid Terminal Oxidase as an Alternative Electron Sink. Plant Physiol. 2008, 149, 1154-1165, doi:10.1104/pp.108.132407.
-
Santos, C.V. Regulation of chlorophyll biosynthesis and degradation by salt stress in sunflower leaves. Scientia Hort. 2004, 103, 93-99, doi:https://doi.org/10.1016/j.scienta.2004.04.009.
-
Wang, P.; Wang, C.-M.; Gao, L.; Cui, Y.-N.; Yang, H.-L.; de Silva, N.D.G.; Ma, Q.; Bao, A.-K.; Flowers, T.J.; Rowland, O.; et al. Aliphatic suberin confers salt tolerance to Arabidopsis by limiting Na+ influx, K+ efflux and water backflow. Plant Soil 2020, 448, 603-620, doi:10.1007/s11104-020-04464-w.
- Krishnamurthy, P.; Mohanty, B.; Wijaya, E.; Lee, D.Y.; Lim, T.M.; Lin, Q.; Xu, J.; Loh, C.S.; Kumar, P.P. Transcriptomics analysis of salt stress tolerance in the roots of the mangrove Avicennia officinalis. Sci Rep 2017, 7, 10031, doi:10.1038/s41598-017-10730-2.
- de Silva, N.D.G.; Murmu, J.; Chabot, D.; Hubbard, K.; Ryser, P.; Molina, I.; Rowland, O. Root Suberin Plays Important Roles in Reducing Water Loss and Sodium Uptake in Arabidopsis thaliana. Metabolites 2021, 11, 735, doi:10.3390/metabo11110735.
Round 2
Reviewer 2 Report
In this revised manuscript, the authors have made extensive changes to improve the original manuscript's quality following the reviewers' suggestions. At this point, I have no more comments. All the issues were properly addressed.
Please, check the suppl. file. The first files of table 2 moved.